# Intent to Be Vaccinated against COVID-19 in Victoria, Australia

**DOI:** 10.3390/vaccines10020209

**Published:** 2022-01-28

**Authors:** Katherine Heath, Aimée Altermatt, Freya Saich, Alisa Pedrana, Stephanie Fletcher-Lartey, Anna L. Bowring, Mark Stoové, Margaret Danchin, Jessica Kaufman, Katherine B. Gibney, Margaret Hellard

**Affiliations:** 1Burnet Institute, 85 Commercial Road, Melbourne, VIC 3004, Australia; aimee.altermatt@burnet.edu.au (A.A.); freya.saich@burnet.edu.au (F.S.); alisa.pedrana@burnet.edu.au (A.P.); stephanie.fletcher@burnet.edu.au (S.F.-L.); anna.bowring@burnet.edu.au (A.L.B.); mark.stoove@burnet.edu.au (M.S.); margaret.hellard@burnet.edu.au (M.H.); 2School of Public Health and Preventive Medicine, Monash University, 533 St Kilda Road, Melbourne, VIC 3004, Australia; 3Department of Paediatrics, University of Melbourne, Parkville, VIC 3010, Australia; margaret.danchin@mcri.edu.au; 4Murdoch Children’s Research Institute, 50 Flemington Road, Parkville, VIC 3052, Australia; jess.kaufman@mcri.edu.au; 5Department General Medicine, The Royal Children’s Hospital Melbourne, Parkville, VIC 3052, Australia; 6Faculty of Medicine, Dentistry and Health Sciences, The University of Melbourne, Parkville, VIC 3010, Australia; 7The Peter Doherty Institute for Infection and Immunity, Department of Infectious Diseases, Melbourne Medical School, University of Melbourne, 792 Elizabeth Street, Melbourne, VIC 3000, Australia; katherine.gibney@unimelb.edu.au; 8Department of Epidemiology and Preventive Medicine, Monash University, 533 St Kilda Road, Melbourne, VIC 3004, Australia; 9Doherty Institute and School of Population and Global Health, University of Melbourne, Parkville, VIC 3010, Australia; 10Department of Infectious Diseases, The Alfred Hospital, 55 Commercial Road, Melbourne, VIC 3004, Australia

**Keywords:** vaccine hesitancy, COVID-19 vaccination, mandatory vaccination, longitudinal cohort, COVID-19 attitudes

## Abstract

Background: High vaccine uptake requires strong public support, acceptance, and willingness. Methods: A longitudinal cohort study gathered survey data every four weeks between 1 October 2020 and 9 November 2021 in Victoria, Australia. Data were analysed for 686 participants aged 18 years and older. Results: Vaccine intention in our cohort increased from 60% in October 2020 to 99% in November 2021. Vaccine intention increased in all demographics, but longitudinal trends in vaccine intention differed by age, employment as a healthcare worker, presence of children in the household, and highest qualification attained. Acceptance of vaccine mandates increased from 50% in October 2020 to 71% in November 2021. Acceptance of vaccine mandates increased in all age groups except 18–25 years; acceptance also varied by gender and highest qualification attained. The main reasons for not intending to be vaccinated included safety concerns, including blood clots, and vaccine efficacy. Conclusion: COVID-19 vaccination campaigns should be informed by understanding of the sociodemographic drivers of vaccine acceptance to enable socially and culturally relevant guidance and ensure equitable vaccine coverage. Vaccination policies should be applied judiciously to avoid polarisation.

## 1. Introduction

Once vaccine supply is secured and the logistics of distribution ensure adequate access, vaccine uptake depends upon strong public support, acceptance, and willingness to be vaccinated [1].

In Australia, attitudes to COVID-19 vaccination have fluctuated during the pandemic. Across various population surveys, vaccine intention reached 84% in April 2020 [2], before declining to 65% in May 2020 and 56% in November 2020 [3]. More recent surveys indicated an increase in vaccine intention to 73% in June 2021 [4], with vaccine hesitancy, including those unwilling and refusing to be vaccinated, at 11.5% in November 2021 [5]. The main reasons for not yet having had a COVID-19 vaccine in Australia have consistently been concerns about side-effects and efficacy [4].

In response to extended outbreaks of the COVID-19 Delta variant in New South Wales and Victoria in mid-2021, policy changes facilitated vaccine uptake, which is now central to long-term management of COVID-19 [6]. For example, in July 2021, Vaxervia (AstraZeneca), previously unavailable to people aged under 60 years, was made available to adults aged 18 years and over in outbreak settings [7], and vaccine uptake in key groups was encouraged through vaccination blitzes [8]. As of 9 November 2021, 90% of adults aged 16 years and over nationally and 92% in Victoria had received at least one COVID-19 vaccine dose.

However, uneven vaccine uptake by socioeconomic status, geography, and social groupings remains a concern [9]. Addressing community concerns regarding COVID-19 vaccination is crucial to build sustained vaccine confidence and high vaccine uptake in Australia, particularly with the call for boosters and vaccination of younger children.

While many existing surveys are serial cross-sectional, reliable monitoring of changing attitudes to vaccination requires prospective cohort surveys. We describe longitudinal changes in vaccine attitudes within a single cohort, which used targeted sampling of priority groups including healthcare workers and people with chronic health conditions to consider attitudes amongst critical occupations and groups.

## 2. Methods

### 2.1. Study Design

The Optimise Study is a multi-disciplinary research platform aiming to quantify government policy efficacy, describe impacts of COVID-19 interventions, and inform adaptive strategies for COVID-19 management [10]. The study was approved by the Alfred Health Ethics Committee (approval number: 333/20). Informed consent was obtained from participants prior to enrolment. The study is ongoing as of November 2021.

The Optimise Study follows a longitudinal cohort of participants and their social networks in the Australian state of Victoria. Recruitment is continuous, with the first participants recruited in September 2020. Participants are intentionally oversampled from key groups considered at risk of (i) contracting COVID-19, (ii) developing severe COVID-19, or (iii) experiencing unintended consequences of COVID-19 restrictions. These groups include employees in aged/health care, older and younger adults, and people from culturally and linguistically diverse backgrounds. Participants are recruited through social media, flyers circulated in community and industry groups, community-based organisations, and professional networks.

### 2.2. Data Collection

Surveys are completed online using NetCollect Survey Software: a data collection software purposively designed to collect and manage social network data. All participants complete a baseline survey and are invited to complete one follow-up survey and four daily contact diaries every four weeks. Participants receive vouchers as reimbursement (AUD$50 for baseline and AUD$35 for active participation in each subsequent month). Details of the survey questions and responses analysed in this study are shown in Table 1. The questions were adapted from existing COVID-19 studies in Australia [11].

### 2.3. Data Analysis

A binary variable was created denoting vaccine intention from Q1 with ‘definitely yes’ = 1, ‘I have already been vaccinated’ = 1, and all other responses = 0. Combination of ‘definitely yes’ and ‘I have already been vaccinated’ into a single outcome variable denoting vaccine intention assumed that vaccine adoption was an appropriate indicator of having or having had vaccine intention. This assumption was supported by examination of longitudinal changes in individual survey responses. A total of 97% of vaccinated participants had previously responded ‘definitely yes’ or ‘probably yes’ to Q1 at least once, demonstrating that, in most instances, vaccination occurred after a clear statement of intent. Further breakdown of responses to Q1 prior to vaccination is given in Appendix A.

A binary variable was created denoting acceptance of vaccine mandates from Q3 with ‘totally acceptable’ = 1 and all other responses = 0.

χ2 testing was used to assess dependency between demographic variables and (a) vaccine intention and (b) acceptance of vaccine mandates in each four-week period.

Binomial-distributed Generalised Estimating Equations (GEEs) with a logit-link function explored differences in longitudinal trends of (a) vaccine intention and (b) acceptance of vaccine mandates between demographic groups. An autoregressive (AR(1)) correlation structure accounted for repeated samples, with decreasing correlation between farther time periods. Participants were clustered by their survey number. Time (in four-week blocks) was included as a continuous covariate. Analysis was conducted using R v.3.6.3 [12] with the geepack package [13].

Model selection used the Quasilikelihood under the Independence Model Criterion (QIC) [14]. Separate models were fit for every combination of demographic fixed effects (age, region of residence at baseline, gender identity, existence of a chronic health condition, country of birth, language spoken at home, whether employed as a healthcare worker at baseline, religion, level of education, employment status, and whether children lived in the household). Models with a QIC in the lowest 10% of all models were subset and, in the interest of model parsimony and to avoid overfitting, the model with fewest covariates in this subset was selected. Interaction terms between time and demographics in the selected model were systematically tested for using the QIC.

## 3. Results

The last four-week period with ≥25 participants unvaccinated or not ‘definitely’ intending to vaccinate was 25 July 2021 to 20 August 2021. Therefore, no χ2 testing was performed using vaccine intention data collected after 20 August 2021 due to insufficient sample size. Barriers to vaccine uptake were not considered after 20 August 2021 for the same reason. There were ≥25 respondents for all time periods and both response categories on acceptance of vaccine mandates, including the most recent data (14 October 2021 to 9 November 2021).

### 3.1. Participant Characteristics

As of 9 November 2021, the study had recruited 726 participants, 686 of whom had complete data for all demographics of interest and had at least one response to both Q1 and Q3. The below analyses were conducted on these 686 participants. Participant demographics are shown in Appendix A.

### 3.2. Vaccine Intention

Among participants who completed a survey between 1 October 2020 and 27 October 2020, 60% reported they would ‘definitely’ get vaccinated against COVID-19, if doses were available. By 14 October 2021 to 9 November 2021, 99% said they would ‘definitely’ get vaccinated or were already vaccinated with at least one dose. It should be noted that these proportions apply to participants that responded during the specified survey period. Not all participants provided a response for every survey period.

Using χ2 testing, vaccine intention differed significantly from 25 July 2021 to 20 August 2021 by age, employment as a healthcare worker, level of education, whether children were living in the household, existence of a chronic health condition, and pre-COVID-19 employment status. χ2 results for every cross-sectional time period are shown in Appendix A.

Model selection on GEEs for vaccine intention designated age, employment as a healthcare worker, level of education, and whether children lived in the household as covariates, in addition to time (in four-week blocks). Interaction terms with time were applied to age, employment as a healthcare worker, and whether children lived in the household. All GEE coefficients and *p*-values for vaccine intention are given in Appendix A.

GEEs indicated that vaccine intention increased over time for all demographics included. Participants aged 35–44 had the lowest vaccine intention in October 2020, although vaccine intention in participants aged 35–44 increased more rapidly than other age categories except age 65 years and over. Healthcare workers had lower vaccine intention than non-healthcare workers in October 2020, but intention increased more rapidly than in non-healthcare workers. Participants with children in the household had lower vaccine intention than those without in October 2020, but intention increased more rapidly over time than in participants without children. Overall, participants with postgraduate qualifications had the highest vaccine intention, and those with Technical and Further Education (TAFE) or trade qualifications had the lowest, although by November 2021, vaccine intention was comparable in all education subgroups.

Figure 1 shows aggregated and disaggregated longitudinal trends in vaccine intention for covariates indicated as significant by GEE model selection. Appendix A shows longitudinal trends in vaccine intention for all other demographic covariates.

### 3.3. Barriers to Vaccine Uptake

Between 25 July 2021 to 20 August 2021, general vaccine safety concerns were the most common reason for not being vaccinated or intending to be vaccinated (56%). The second biggest concern was blood clots (24%), followed by vaccine efficacy concerns (22%). Longitudinal trends in reasons for not intending to receive a COVID-19 vaccine are shown in Figure 2.

### 3.4. Vaccine Mandates

Among participants who completed a survey between 14 October 2021 and 9 November 2021, 71% responded that mandatory vaccination for high-risk groups was ‘totally acceptable’, an increase from 50% between 1 October 2020 and 27 October 2020. Unvaccinated participants or participants not ‘definitely’ intending to vaccinate were less likely to accept vaccine mandates between 25 July 2021 and 20 August 2021 (*p* < 0.01, χ2).

Using χ2 testing, acceptance of vaccine mandates significantly differed between 14 October 2021 and 9 November 2021 by age, gender, level of education, existence of a chronic health condition, country of birth, language, pre-COVID-19 employment status, and whether children lived in the household. χ2 results for every cross-sectional time period are shown in Appendix A.

Model selection on GEEs for acceptance of vaccine mandates designated age, gender, and level of education as covariates, in addition to time (in four-week blocks). An interaction term with time was applied to age. All GEE coefficients and *p*-values for acceptance of vaccine mandates are given in Appendix A.

GEEs indicated that acceptance of vaccine mandates increased over time for all demographics except participants aged 18–24 years, where it decreased. However, participants aged 18–24 were the most accepting in October 2020 and participants aged 35–44 were the least accepting. Acceptance increased most rapidly in participants aged 55–64 years. Participants identifying as female had higher acceptance of vaccine mandates than those identifying as male. Participants with a primary education were the most accepting, and those with a TAFE or trade qualification were the least accepting.

Figure 3 shows aggregated and disaggregated longitudinal trends in acceptance of vaccine mandates for covariates indicated as significant by GEE model selection. Appendix A shows longitudinal trends in acceptance of vaccine mandates for all other demographic covariates.

## 4. Discussion

Willingness to be vaccinated is critical to the success of national COVD-19 vaccination rollouts. A key result from this study is the increase in vaccine intention in Victoria, Australia, since October 2020, despite fluctuation and variation across key demographics.

A limitation of our study is that vaccine intention does not capture the motivations for vaccine uptake. Individuals may have polarised opinions about COVID-19 vaccination in general that are not reflected in their intentions. COVID-19 vaccine willingness must be sustained to tackle new variants and administer boosters. Whilst our study has described a rapid initial increase in vaccine intention, further studies would be well placed to consider the complex motivations and beliefs surrounding COVID-19 vaccination to assess sustained community engagement.

Internationally, COVID-19 vaccine intention has been highly variable between countries, and the determinants of vaccine intention have appeared complex. Several studies have highlighted associations between vaccine intention and political ideology, government trust, and perceptions of a government’s handling of the COVID-19 pandemic [15,16,17]. However, several outlier countries have been noted. A belief that the government is handling the pandemic well was negatively associated with vaccine intention in Brazil and the United States but was positively associated in other countries [15]. Trust in the government has been associated with intention to vaccinate, except in Japan [18]. Australia, along with Norway and China, has been observed to have high COVID-19 vaccine intention compared to other countries [18,19].

Increased vaccine intention in our study likely arose from a combination of factors. Outbreaks of the COVID-19 Delta variant in New South Wales and Victoria in mid-2021 prompted a shift in Victoria’s COVID-19 response strategy from suppression to co-existence. This shift was synonymous with a need for high vaccination coverage to reduce transmission. Other motivating factors may include desire to travel, healthcare worker recommendation, and the introduction of widespread vaccine mandates in several workplaces including hospitality and retail.

The Optimise Study is uniquely positioned to detect longitudinal trends in demographic groups due to oversampling of key groups in a longitudinal cohort.

We found that healthcare workers had higher vaccine intention than the general population since December 2020. Healthcare workers play a crucial role in public decision making and may affect vaccine uptake in the general population [20]. Participants aged 65 years and over had higher vaccine intention than other age groups, consistent with previous studies in Australia and overseas [5,21]. This age group had earlier access to COVID-19 vaccines in Australia and has the greatest risk of morbidity and mortality from COVID-19 [22]. Higher vaccine intention in older age groups has been common in other countries in North America and Europe [19], although some outlier countries, including Mexico, have observed higher vaccine intention in younger age groups [23].

Participants with children in their household had lower vaccine intention than those without, although intention in the two groups became equivalent by November 2021. Pre-pandemic vaccine hesitancy in relation to childhood vaccines is well-documented [24] and may influence COVID-19 vaccination intention. Countries with low pre-pandemic childhood vaccine confidence such as the United States and Russia have been observed to have lower COVID-19 vaccine acceptance [25]. Across six countries in Europe, North America, and Asia, caregivers were more likely to vaccinate their children against COVID-19 if they were up-to-date with other childhood vaccinations [26]. Parents’ hesitancy about COVID-19 vaccines for themselves may be due to increased exposure to vaccine skepticism [27] and heightened concerns about vaccine safety [24].

Vaccine uptake was consistently slightly higher in our cohort (99% reporting at least one dose between 14 October 2021 and 9 November 2021) than in the general population in Victoria (92% on 9 November 2021) [28]. See Appendix A for a longitudinal comparison. Oversampling of key groups in our cohort increased power to compare longitudinal trends between demographics; however, overall cohort summary statistics may not always represent the general population. Disaggregated statistics from other studies corroborate our results; in particular, greater vaccine intention has been associated with age and level of education [5,29]. However, few studies in Australia have presented data beyond mid-2021.

Acceptance of vaccine mandates in our cohort increased most rapidly in participants aged 55–64 and 65 and over and was lowest for participants aged 18–24. Young people may be disproportionately impacted by mandates, for example through mandates on industries with predominantly younger workers such as hospitality or greater financial insecurity entailing limited choice if an employer enforces a mandate. Participants with chronic health conditions were more accepting of vaccine mandates than those without since April 2021, implying greater acceptance in those that mandates protect from serious illness.

In 2020, respondents in 19 countries including Brazil, Russia, India, China, the United States, and the United Kingdom reported they would be less accepting of a COVID-19 vaccination if it was mandated by an employer [17]. In our cohort, participants without vaccine intention were less likely to accept vaccine mandates. Opinions on vaccine mandates likely reflect a multifaceted interaction of social, economic, political, and cultural drivers as well as the degree of trust in authority and value placed on personal autonomy.

In overseas studies, childhood vaccine mandates have caused socio-political polarisation [30]. In Australia, polarisation concerning COVID-19 vaccines is stronger than previously observed regarding childhood vaccination [31]. Vaccine hesitancy has been found to fall along ideological divides, with individuals’ moral foundations being associated with their degree of vaccine hesitancy [32]. Communication and community engagement on COVID-19 and vaccination should acknowledge different social and cultural contexts associated with vaccine apprehension.

Vaccine intention fluctuated over time and between demographics in our cohort. Vaccine safety concerns remained firm in participants without vaccine intention. Ongoing concerns must be addressed given the need for booster vaccines to reduce waning immunity and vaccination for younger children. Community engagement and participation in health messaging may further improve uptake [9]. It is critical that there is accessible, evidence-informed policy that is applied judiciously and with consideration of cultural differences to maintain and increase vaccine confidence and strengthen COVID-19 vaccination programmes both in Australia and internationally.

## Figures and Tables

**Figure 1 vaccines-10-00209-f001:**
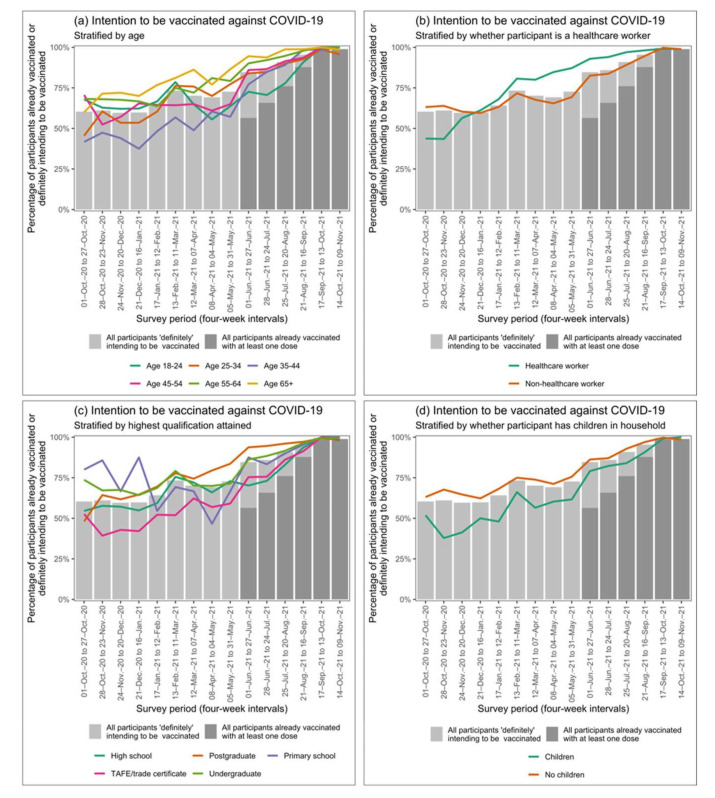
Longitudinal trends in vaccine intention in demographic groups. Grey bars display vaccine intention in all survey respondents, separated by those responding ‘Definitely yes’ and ‘I have already been vaccinated’ to Q1 (see Table 1 for survey question specification). Note that data on vaccine uptake are presented from 1 June 2021 onwards only due to the introduction of survey questions assessing vaccine status in May 2021 (see Table 1). Lines display vaccine intention disaggregated by (**a**) age, (**b**) employment as a healthcare worker, (**c**) highest education qualification attained, and (**d**) whether children were present in the household. This figure displays longitudinal trends for the demographics elected by Generalised Estimating Equation (GEE) model selection only. See Appendix A for longitudinal trends in other demographic groups.

**Figure 2 vaccines-10-00209-f002:**
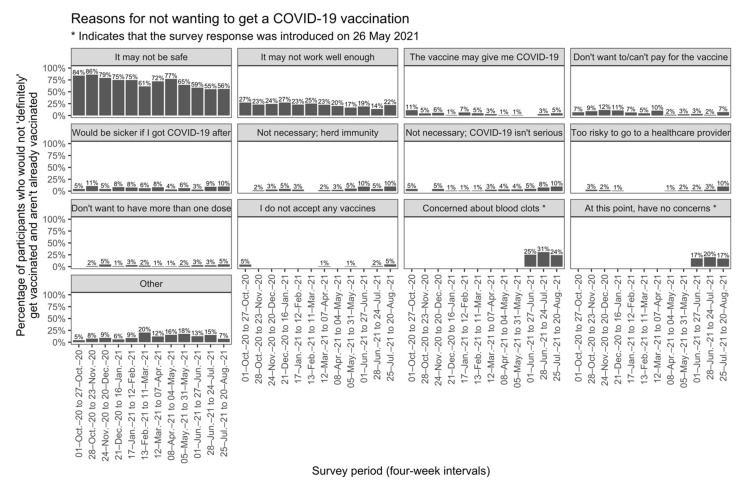
Reasons for not receiving a COVID-19 vaccination in participants who did not respond ‘definitely yes’ or ‘already vaccinated’ when asked in Q1 if they would be vaccinated. Participants were able to select all reasons that applied to them. The most frequent ‘other’ reasons were uncertainty about pre-existing medical conditions or medications interacting with vaccines; concerns about blood clots or desire to choose a vaccine; the opinion that there is no urgency to be vaccinated in Australia; and concerns about effects of vaccination on pregnancy, breastfeeding, or fertility. Responses were examined only for months with ≥25 respondents who did not respond ‘Definitely yes’ or ‘I have already been vaccinated’ to Q1 (see Table 1 for survey question specification). * Survey response was introduced on 26 May 2021 and data from 5 May 2021 to 31 May 2021 are omitted because not all participants in this period were asked the question.

**Figure 3 vaccines-10-00209-f003:**
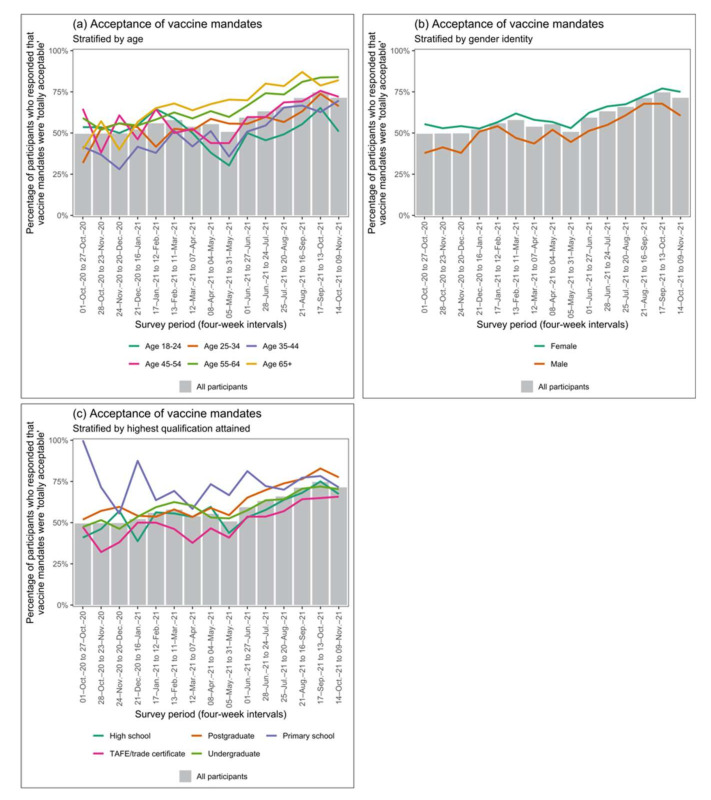
Longitudinal trends in acceptance of vaccine mandates in demographic groups. Acceptance of vaccine mandates was considered present in participants who responded that vaccine mandates for high-risk groups were ‘totally acceptable’ (see Table 1 for survey question specification). Grey bars display acceptance of vaccine mandates in the aggregated survey population. Lines display acceptance of vaccine mandates disaggregated by (**a**) age, (**b**) gender identity, and (**c**) highest education qualification attained. This figure displays longitudinal trends for the demographics elected by Generalised Estimating Equation (GEE) model selection only. See Appendix A for longitudinal trends in other demographic groups.

**Table 1 vaccines-10-00209-t001:** Details of survey questions asked to participants. Participants were invited to complete surveys every four weeks. On 26 May 2021, Q1 and Q2 were revised. Q1 (v1) was asked prior to 26 May 2021 and Q1 (v2) was asked after 26 May 2021. Two additional responses were added to Q2 on 26 May 2021, indicated by an asterisk.

	Question	Conditional Logic	Possible Responses
Q1 (v1)	If a COVID-19 vaccine was to become available to everyone in Australia, do you think you would have it yourself?	None	Definitely yes
			Probably yes
			Probably not
			Definitely not
			Unsure
Q1 (v2)	Do you think you would have a COVID-19 vaccine?	None	I have already been vaccinated
			Definitely yes
			Probably yes
			Probably not
			Definitely not
			Unsure
Q2	For what reason(s) would you not have a COVID-19 vaccine yourself? Select all that apply.	Q2 asked only to participants who answered anything other than ‘definitely yes’ or ‘I have already been vaccinated’ to Q1	It will not be needed as most people will have had the infection by then
			I don’t think the vaccine is necessary because COVID-19 is not that serious in most people
			It may not work well enough to be worth having
			I am worried that it is not safe and hasn’t been tested enough for safety
			I am worried that I might catch COVID-19 from the vaccine
			I am worried that I would get sicker if I got COVID-19 after the vaccine
			I do not want to/am not able to pay for the vaccine
			I do not want the vaccine if there is more than one dose
			I do not want to attend a health care provider to have the vaccine due to the risk of catching COVID-19
			I do not accept any vaccines for myself so would not accept a COVID-19 vaccine
			Other
			At this stage I have no concerns about the vaccine *
			I am worried that I may develop a blood clot after getting the COVID-19 vaccine *
Q3	How acceptable do you think it is to have a mandatory COVID-19 vaccine for certain high-risk groups, such as healthcare workers, given the current COVID-19 pandemic in Australia?	None	Highly acceptable
			Somewhat acceptable
			Neutral
			Somewhat unacceptable
			Totally unacceptable

* Survey response was introduced on 26 May 2021.

## Data Availability

The data presented in this study may be requested from the corresponding author. The data are not publicly available due to ethical considerations and data privacy restrictions.

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
