# Peer review of "Intent to Be Vaccinated against COVID-19 in Victoria, Australia"

_vaccines, 2022, doi:10.3390/vaccines10020209_

Round 1

Reviewer 1 Report

This is potentially a very interesting and useful study. One major limitation is that that outcome variable is not "intention" per se. It is a combination of intention and action (i.e. the actual behavior of adopting a vaccine). This muddies the water. It is not clear what proportion of the outcome comprises of vaccine adopters versus vaccine intenders. 

I would recommend that that the authors remove vaccine adopters from their sample and limit the analysis to vaccine intenders.  This will give a much clearer picture of what actually happened to respondents' intentions to get vaccinated.  

Can the authors provide some information as to whether they think this sample is representative at all of general trends in Australia? This may be difficult to do but, if there are other studies which show similar or different trends, an argument could possibly be made one way or the other.

Author Response

Response to Reviewer 1 Comments

Point 1: This is potentially a very interesting and useful study. One major limitation is that that outcome variable is not "intention" per se. It is a combination of intention and action (i.e. the actual behavior of adopting a vaccine). This muddies the water. It is not clear what proportion of the outcome comprises of vaccine adopters versus vaccine intenders. 

I would recommend that that the authors remove vaccine adopters from their sample and limit the analysis to vaccine intenders.  This will give a much clearer picture of what actually happened to respondents' intentions to get vaccinated.  

Response 1: Thank you for the comment on the combination of intention and action. It certainly prompted a lot of complex conceptual thinking about the research question on my part, which was very valuable. I summarise my response in three main points:

  1. The proportion of vaccine adopters versus intenders has been added.

We agree with the reviewer that it will certainly assist the interpretation of the results to report the proportion of the outcome variable that are adopters versus intenders. We have now demonstrated this in our figures both in the main text and supplementary material. Note that this is only possible from 01-Jun-2021 onwards, due to the question assessing vaccination status being introduced as an addendum to the survey, as noted in Table 1. We have noted this in the figure captions.

  1. Removal of vaccine adopters would not be a viable analysis option.

I note two main reasons that removal of vaccine adopters would not be suitable.

  • Removal of vaccine adopters would have detrimental effects on sample size. By November 2021, 99% of respondents were vaccinated with at least one dose. If all vaccine adopters were removed from the sample, only 1% of the cohort would remain; considerable data loss. Problems also occur if the responses of participants are omitted after the point that they become vaccinated. If this were done, the sample size would become gradually smaller as vaccine uptake increases.
  • Removal of vaccine adopters would give an inaccurate representation of vaccine attitudes in the general population. If the survey responses of vaccine adopters were omitted after the point that they report being vaccinated, then the ratio of participants not intending to vaccinate to participants intending to vaccinate would either stay the same or increase. This would occur because vaccine intenders would be moving to the vaccine adopters category and thereby removed from analysis. Such a ratio would imply an increase in vaccine rejection, which would be inaccurate and an artefact of removing vaccine adopters.

I appreciate the reviewer’s perspective, that having being vaccinated may not be precisely synonymous with intent to vaccinate. However, the removal of vaccine adopters from the dataset would lead to more confusion in the results for the reasons highlighted above.

  1. I argue that the combination of vaccine intenders and adopters is justifiable.

Our analysis assumes that vaccine adoption implies vaccine intention. A person who has vaccinated themselves has expressed their intent to do so by consenting to get the vaccine. Of course, the motivations for intent to vaccinate can be hugely diverse: concern for personal health, concern for vulnerable relatives, desire to travel, employment in a high-risk setting, compliance with mandates, etc. However, the objective of this study is not to quantify the motivations for vaccination but is to quantify intention to vaccinate.

From the above, I argue that vaccine intention is a necessary but insufficient condition for vaccine adoption. Vaccine intention will not always entail adoption, for example due to geographic and demographic variability in vaccine access. However, someone who has been vaccinated can be inferred to have done so through intent, although the reasons for this intent may be highly heterogeneous.

Therefore, the combination of vaccine adopters and intenders is justifiable to quantify vaccine intent because those that have been vaccinated will also have/have had vaccine intent. The reason we focus on intent and not adoption is that adoption will be more heavily influenced by vaccine eligibility, thereby more greatly confounding any association with demographics. Focus on intent also enables longitudinal analysis of data prior to vaccine rollout.

To demonstrate the association between vaccine intent and vaccine adoption, I have examined longitudinal changes in individual participants’ survey responses. Only 3% of vaccinated participants got vaccinated without having previously responded at least once that they would ‘definitely’ or ‘probably’ get vaccinated. Therefore, for 97% of vaccinated participants, vaccine adoption occurred after an explicit statement of intention to vaccinate themselves. I have now included these statistics in the supplementary material (supplementary table 1) and have added a clause in the methods of the main text clarifying our assumptions.

I acknowledge that there is likely a distinction between intention to vaccinate and desire to vaccinate. However, assessing this would a complex question that would probably warrant its own specialised survey and may be out of the scope of this manuscript. I have added a section to the discussion highlighting the distinction between intention and motivation as a limitation of the study and recommending it as an area for further research.

Point 2: Can the authors provide some information as to whether they think this sample is representative at all of general trends in Australia? This may be difficult to do but, if there are other studies which show similar or different trends, an argument could possibly be made one way or the other.

Response 2: I have added a paragraph to the main text discussion and a figure to the supplementary material (Figure S3) comparing the vaccine coverage in our cohort to the actual vaccination coverage in Victoria as published by the Australian Government Department of Health. Because our cohort intentionally oversamples key groups to increase power to compare demographics, vaccine coverage was higher in our cohort than in the general population, although the trend was comparable.

Existing studies have certainly shown similar associations between vaccine intention/willingness in Victoria and the demographics we have identified, particularly by age and level of education (see Melbourne Institute and Biddle et al., references 5 and 29 in main text), and these are noted in the discussion. However, few studies have presented data – particularly data disaggregated by our demographics of interest - beyond mid-2021, making comparison of our data to existing trends in demographic subgroups very challenging. This is certainly an area where new studies would be beneficial.

Also note that general trends in vaccine intention in Australia may be heterogeneous between states due to differences in state-level policies on vaccine accessibility, mandates and different levels severity of historical COVID-19 outbreaks. I have indicated in the main text and Supplementary Figure S3 that comparisons of cohort data to Australian Government Department of Health data are for Victoria only.

Reviewer 2 Report

The authors report a longitudinal study of vaccine attitudes within a single cohort of Australian respondents and using targeted sampling over a period of just over 12 months, starting in October 2020. A simple and very short survey is used and data over time are reported and discussed.

The study is within the scope of the journal and it will certainly be of interest to the readership – indeed it will be of global interest to both scientists and professional working in public health. The major strength of the study is its longitudinal nature, which allows data within participants to be linked over a rather long period of time, thus complementing the numerous cross-sectional studies of COVID vaccination acceptability that have been published to-date. I do not have any major reservations about the study design or reporting; I would suggest that the study is also reviewed by a methodologist/statistician to check the approach to the data analysis, on which I’m not able to comment in detail. A few suggestions for the authors to consider, as follows:

The results section in the abstract is rather vague; can you add some numerical results, to give an immediate picture of the acceptability rates during the course of the study (as an example)?

The discussion section is quite thin and rather descriptive. I would strongly recommend better coverage of what other studies, globally, have shown during the same timeline in terms of COVID vaccination uptake – there are by now reviews of such studies that have started to appear in the literature. The focus on Australia is to be expected, but the study currently does not seem ‘outward-looking’ enough for an international reader (like me). Further to this, have there been any major national or other campaigns within Australia during the timeline of the study that a reader needs to be aware of, as they would provide context for the graphs/slopes you have reported. A para with a description of what the public health policies have been would be very useful to offer such context. Lastly, make sure to include a paragraph at least on the limitations of the study.

Lastly: I appreciate this may be beyond the scope of the study as such, however it seems to me that there would be value in descriptively linking your (overall) results in the discussion with the actual vaccination coverage achieved in Australia during the same time period. Do the % values seem to follow a similar pattern, eg in terms of the overall vaccination uptake and/or the slope of the uptake curve? I appreciate the data cannot be linked as such, but as we have so much publicly available data on coverage it seems an omission to not mention them at all in reflecting on what you have found.

Author Response

Response to Reviewer 2 Comments

The authors report a longitudinal study of vaccine attitudes within a single cohort of Australian respondents and using targeted sampling over a period of just over 12 months, starting in October 2020. A simple and very short survey is used and data over time are reported and discussed.

The study is within the scope of the journal and it will certainly be of interest to the readership – indeed it will be of global interest to both scientists and professional working in public health. The major strength of the study is its longitudinal nature, which allows data within participants to be linked over a rather long period of time, thus complementing the numerous cross-sectional studies of COVID vaccination acceptability that have been published to-date. I do not have any major reservations about the study design or reporting; I would suggest that the study is also reviewed by a methodologist/statistician to check the approach to the data analysis, on which I’m not able to comment in detail. A few suggestions for the authors to consider, as follows:

Point 1: The results section in the abstract is rather vague; can you add some numerical results, to give an immediate picture of the acceptability rates during the course of the study (as an example)?

Response 1: Thank you for the comment. We have added some numerical descriptions of the increases in vaccine intention and acceptance of vaccine mandates to the abstract.

Point 2: The discussion section is quite thin and rather descriptive. I would strongly recommend better coverage of what other studies, globally, have shown during the same timeline in terms of COVID vaccination uptake – there are by now reviews of such studies that have started to appear in the literature. The focus on Australia is to be expected, but the study currently does not seem ‘outward-looking’ enough for an international reader (like me).

Response 2: I agree with the reviewer that there was room in the manuscript to put the results of our study into a global context. I have added a paragraph highlighting the general determinants of vaccine intention that have been observed in several countries and have pointed to some outliers. I have also put several of the more specific results of our study into context by comparing them with results in other countries. Please see track changes in the discussion for these changes and references.

Point 3: Further to this, have there been any major national or other campaigns within Australia during the timeline of the study that a reader needs to be aware of, as they would provide context for the graphs/slopes you have reported. A para with a description of what the public health policies have been would be very useful to offer such context.

Response 3: Thank you for the very valid comment. I have added a section to the introduction, which describes and cites two main policy changes that may have influenced vaccine intention during the study. I acknowledge that there were other policies that may have been influential and my description may not be exhaustive. In response extended outbreaks of the COVID-19 Delta variant in Victoria and New South Wales in mid-2021, adults under 60 years were able to access the AstraZeneca vaccine for the first time due to (i) supply shortages of the Pfizer vaccine and (ii) the fact that the risk of blood clots was outweighed by the risks of COVID-19 in outbreak settings. In addition, vaccine blitzes were undertaken in key populations throughout Victoria and New South Wales. It is quite visible in the figures that from June 2021 onwards there was a large uptick in vaccine intention and uptake, which coincides with these changes in policy. These policy changes, as now cited in the introduction, give context to the observed increase in vaccine intention.

Point 4: Lastly, make sure to include a paragraph at least on the limitations of the study.

Response 4: I have added two paragraphs to the discussion that speak to limitations. First, I discuss the difficulties in comparing our results to population-level statistics in Victoria (see my response to the proceeding comment for further detail on this). Second, I discuss the distinction between intention and motivation to be vaccinated, noting that our study cannot distinguish the complex reasons for which people intend to vaccinate themselves. This will become increasingly important as long-term community engagement is needed for booster shots and in response to new variants.

Point 5: Lastly: I appreciate this may be beyond the scope of the study as such, however it seems to me that there would be value in descriptively linking your (overall) results in the discussion with the actual vaccination coverage achieved in Australia during the same time period. Do the % values seem to follow a similar pattern, eg in terms of the overall vaccination uptake and/or the slope of the uptake curve? I appreciate the data cannot be linked as such, but as we have so much publicly available data on coverage it seems an omission to not mention them at all in reflecting on what you have found.

Response 5: Thank you for pointing this out; I agree it would be valuable to directly compare vaccine attitudes and uptake in our study with the actual vaccination coverage achieved in Australia. I have accessed uptake data published by the Victorian government and present a comparison in the supplementary material (Figure S3), as well as a discussion point in the main text.

I list three caveats that are also now noted in the main text:

  1. Data from the Australian government on vaccine uptake describes coverage in adults aged 16+, whilst our study recruited adults aged 18+. It is unfortunate that there is no way to disaggregate the government data to make a direct comparison. Therefore, it might be expected that vaccine uptake in our cohort is higher than the general population as vaccines were only made available to younger people more recently.
  2. Our study purposefully oversamples key groups to facilitate comparisons between demographics (including older adults or employees in professions with high transmission risk). Due to oversampling of key groups that may have been eligible for vaccination sooner or are more at risk of COVID-19 morbidity/mortality, the vaccine coverage in our cohort without disaggregation may be higher the coverage in the general population.
  3. It should also be noted that general trends in vaccine intention in Australia may be heterogeneous between states due to differences in state-level policies on vaccine accessibility, mandates and different levels severity of historical COVID-19 outbreaks. I have indicated in the main text and Supplementary Figure S3 that comparisons of cohort data to Australian Government Department of Health data are for Victoria only.